# Birth cultures: A qualitative approach to home birthing in Chile

**Pía Rodríguez-Garrido**[1,2], **Josefina Goberna-Tricas**[1]*

1 Department of Public Health, Mental Health and Perinatal Nursing, Faculty of Medicine and Health Sciences, ADHUC Research Centre: Theory, Gender and Sexuality, University of Barcelona, Barcelona, Spain, 2 Department of Health, University of O'Higgins, O'Higgins, Chile

* jgoberna@ub.edu

**Data Availability Statement:** All relevant data are within the paper and its Supporting information files.

## Abstract

### Background

Birth cultures have been transforming in recent years mainly affecting birth care and its socio-political contexts. This situation has affected the feeling of well-being in women at the time of giving birth.

### Aim

For this reason, our objective was to analyse the social meaning that women ascribe to home births in the Chilean context.

### Method

We conducted thirty semi-structured interviews with women living in diverse regions ranging from northern to southern Chile, which we carried out from a theoretical-methodological perspective of phenomenology and situated knowledge. Qualitative thematic analysis was used to analyse the information collected in the field work.

### Findings

A qualitative thematic analysis produced the following main theme: 1) Home birth journeys. Two sub-categories: 1.1) Making the decision to give birth at home, 1.2) Giving birth: (re) birth. And four sub-categories also emerged: 1.1.1) Why do I need to give birth at home? 1.1.2) The people around me don't support me; 1.2.1) Shifting emotions during home birth, 1.2.2) I (don't) want to be alone.

### Conclusion

We concluded that home births involve an intense and diverse range of satisfactions and tensions, the latter basically owing to the sociocultural resistance surrounding women. For this reason, they experienced home birth as an act of protest and highly valued the presence of midwives and their partners.

**Funding:** The research leading to these results has received funding from Spanish Ministry of Science, Innovation and Universities [grant number PGC2018-094463-B-100- funded by MCIU/AEI/FEDER, UE], and too from the National Scientific and Technological Commission of Chile (CONICYT) Scholarship program for Doctorates Abroad, 2017 [grant number 72180224-2017] which has founded the doctoral scholarship of the first author. The funders had no role in study design, data collection and analysis, decision to publish, or preparation of the manuscript.

**Competing interests:** The authors have declared that no competing interests exist.

# Introduction

Birth cultures are the set of traditions and rites handed down from generation to generation concerning a unique event in the lives of human beings: birth. Birth cultures evolve in response to their historical, political, economic and social contexts, which regulate and condition the community in which birth takes place [1]. However, when we speak about birth cultures, we refer not only to the act and moment of childbirth itself, but also to maternity, the gestation process, the post-partum period, breastfeeding, child-raising, connections with birth attendants and interactions with institutions [2, 3].

From this perspective, birth cultures have become a valuable epistemological resource for study. Their transitional nature has led authors to probe deeper since the 1970s [4] and analyse paradigms ranging from biologicist and conventional perspectives to criticism by human behavioural and social sciences that question the ways in which the imaginaries [5] and social representations [6] of birth have been constructed.

## Feminist approaches to motherhood

The feminist movement has contributed to this dialogue by positioning the categories 'woman' and 'mother' in the debate. Equality feminist authors such as Simone de Beauvoir [7] have dismantled the reproductive essentialism imposed upon woman as a subject, indicating that, to her, maternity has meant subordination. In the same vein, Elisabeth Badinter [8] explored how patriarchal society has interpolated 'maternal instinct' as a category to keep women subordinate through self-denial and sacrifice to their children.

Alternatively, difference feminism understands maternity as an identity and a political role. In this respect, Luce Irigaray [9] reclaims the figure of mother insofar as her body acts as the epistemological setting wherein the symbolic new order of the feminine is rewritten in genealogy. Supporting that idea, Adrienne Rich [10] highlights the importance of seeing the body as an experience of maternity to produce knowledge that will generate transcendental societal change.

Today, both women and mothers who experience diverse feminist spaces bear witness to the difficulties of pregnancy, giving birth and motherhood [11]. The dominant structures instated by the neoliberal socio-economic model which guides societal behaviours have violated and entrenched insecurity in the collective dynamics associated with motherhood and parenting. These neoliberal structures have been expressed in part and supported by the biomedical model, which permanently patrols women's bodies through healthcare procedures and protocols [12, 13].

Epistemological perspectives that make it possible to problematise multiple aspects of this debate have emerged. They are an important from of resistance, in part because these perspectives additionally restore and normalise the voices of women-mothers as the protagonist of their stories.

## Pathways and stresses toward home birth

Authors including Casilda Rodrigáñez [14] and Ibone Olza [3] have taken on large challenges from the perspective of a humanist paradigm of health. Olza [3] developed a critical reading of the patriarchal, technocratic model for thinking about birth when she stated that 'giving birth seems problematic if one views women's bodies as an imperfect, defective and poorly-constructed version of the male body, as has traditionally been the view of androcentric medical science' [3].

For her part, Robbie Davis-Floyd [15] considers industrialisation as the historic event that changed cultural behaviours and therefore how childbirth and home births were viewed and

attended to. Michel Odent [16] further describes progressive changes associated with the physiology of birth as the result of hyper-medicalising care provided during labour.

Birthing has come a long way from taking place in the privacy of the home, attended by a birth assistant, to relocation inside hospital wards, a path that has gradually led to the medicalisation of a physiological process [17]. In response to this shift—one that has had significant consequences for women, both physically and psychologically [16–18]—several social collectives have emerged that point to home birthing as a solution to birth-related psychosocial problems [19, 20].

However, home birthing with a midwife as one's healthcare provider is a right that not all women have unfettered access to; this fact is clear from how varied distinct geopolitical, healthcare and economic situation are among countries. While home births enjoy protocols and social security system financing in the Netherlands, England and Australia [21, 22], other countries face a very different reality, particularly Latin American ones such as Chile, where home births are not covered by government regulations, nor is it financed by the national social security system [22, 23]. Despite the situation in Chile, home birthing has become an increasingly popular choice among women and has gradually become part of the national landscape. For this reason, this research project aims to contribute to the literature about Chile, a country where scientific documentation on the subject is lacking, by analysing the social meaning that women there ascribe to home births.

## Method

### 1. Design

The study was designed according a qualitative methodological approach, which was defined by Denzin et al. [24] as 'a situated activity that locates the observer in the world. It consists of a set of interpretive, material practices that make the world visible. These practices transform the world' [24]. This practice therefore facilitates a respectful, situated approach to the reality researched.

On this basis, we selected the phenomenological paradigm as our methodology. From a feminist perspective, the phenomenological paradigm 'emerges from an inter-relational ontology, that not only does it offer the account of embodied experience for which it is usually recognised, but also that embodied perception underlies the production of knowledge and grounds politics' [25].

In this regard, and with feminist epistemologists in mind, Donna Haraway [26] has proposed that situated knowledge is an interpolation of positivist research methods and models. She emphasises the urgency of the partial perspective we might encounter, in our specific case, in the vision of Chilean women who give birth at home, since they are embodied knowledge 'in order to name where we are and are not, in dimensions of mental and physical space we hardly know how to name' [26].

### 2. Context of the study

**Healthcare and birthing in Chile.** Chile is a country located on the southwestern coast of South America. Considered a developing country by the World Bank, Chile has the highest gross domestic product (GDP) per capita in its region [27, 28] and its political and economic histories are marked by the well-known fact that it was the first Latin American country to enact a neoliberal dictatorship during the 1980s. That circumstance paved the way for the privatisation of healthcare, positioning it as a consumer good. A mixed public-private healthcare system was established in which public insurance (*Fondo Nacional de Salud*, *FoNaSa*) and

private insurance (*Instituciones de Salud Previsional*, *ISaPres*) coexist. This system is still in place today [29].

The implementation of this model gave rise to a technocratic, commodified view of healthcare processes, including the healthcare provided during childbirth. According to recent statistics, 99.6% of births in Chile have taken place at a healthcare institution and have been attended by healthcare professionals, namely, OB/GYNs and midwives [30]. Within the public system, 33% of birth are by caesarean delivery and 63% of those in private system are by caesarean, positioning Chile as a country with one of the world's highest rates of C-sectioning [31].

In contrast, the number of home births has increased significantly. In 2017, 0.15% of births took place in the home with an attending healthcare professional [3], while in 2020, approximately 1% of home births were attended by a midwife [32].

**Participant context.**   Chile is divided into 16 administrative regions. The women interviewed for this study live in the regions of Norte Grande ('Far North'), Zona Central ('Central Chile') and Zona Sur ('Southern Zone') (Fig 1).

These regions were selected to enlarge the geographical scope of home births by women in both rural and urban areas of Chile. This aspect an important consideration due to the fact that women described difficulties finding a midwife who was able to attend births in locations far away from urban areas because of the low availability and quantity of midwives in rural areas. Moreover, a mandatory provision within the protocols established by midwives attending home births requires that childbirth take place in a location that is closer than 20 minutes' drive from a hospital in case an emergency transfer becomes necessary.

In light of each of these considerations, Chile's extensive geographical expanse and the diversity in expertise that this breadth occasions means that residence in rural areas or places outside of the capital's centralising sphere of influence is a crucially important circumstance that must be taken into account during participant screening and selection.

## 3. Participant recruitment

Potential participants were identified for this study in two ways. The first was to contact the 'Asociación Gremial de Matronas de Parto en Casa de Chile: Maternas Chile' ['Maternas Chile' Professional association for midwives attending home births] via e-mail. They collaborated with the project by disseminating information among women who had attended births. Ten women who were interested in participating in the study responded to this initial call.

The second method used social media, specifically Facebook. A notification posted on this platform invited women who wanted to share their home birth experiences to participate in the study. Approximately 100 women responded to this second strategy for identifying participants, a number far exceeding the first strategy and strongly contrasting the presumed difficulty of finding participants.

Both recruitment strategies were carried out from May to August of 2018. Initial conversations with possible study participants took place via e-mail and WhatsApp during these months.

## 4. Participant selection and characteristics

Because recruitment attracted participants through the *Asociación Gremial de Matronas de Parto en Casa de Chile 'Maternas Chile'* ['Maternas Chile' Professional association for midwives attending home births] and, especially, through Facebook, the following participant selection criteria were applied: Chilean women who planned a home birth that was carried out with a midwife, who were legally adults when they gave birth and at the time of the interview,

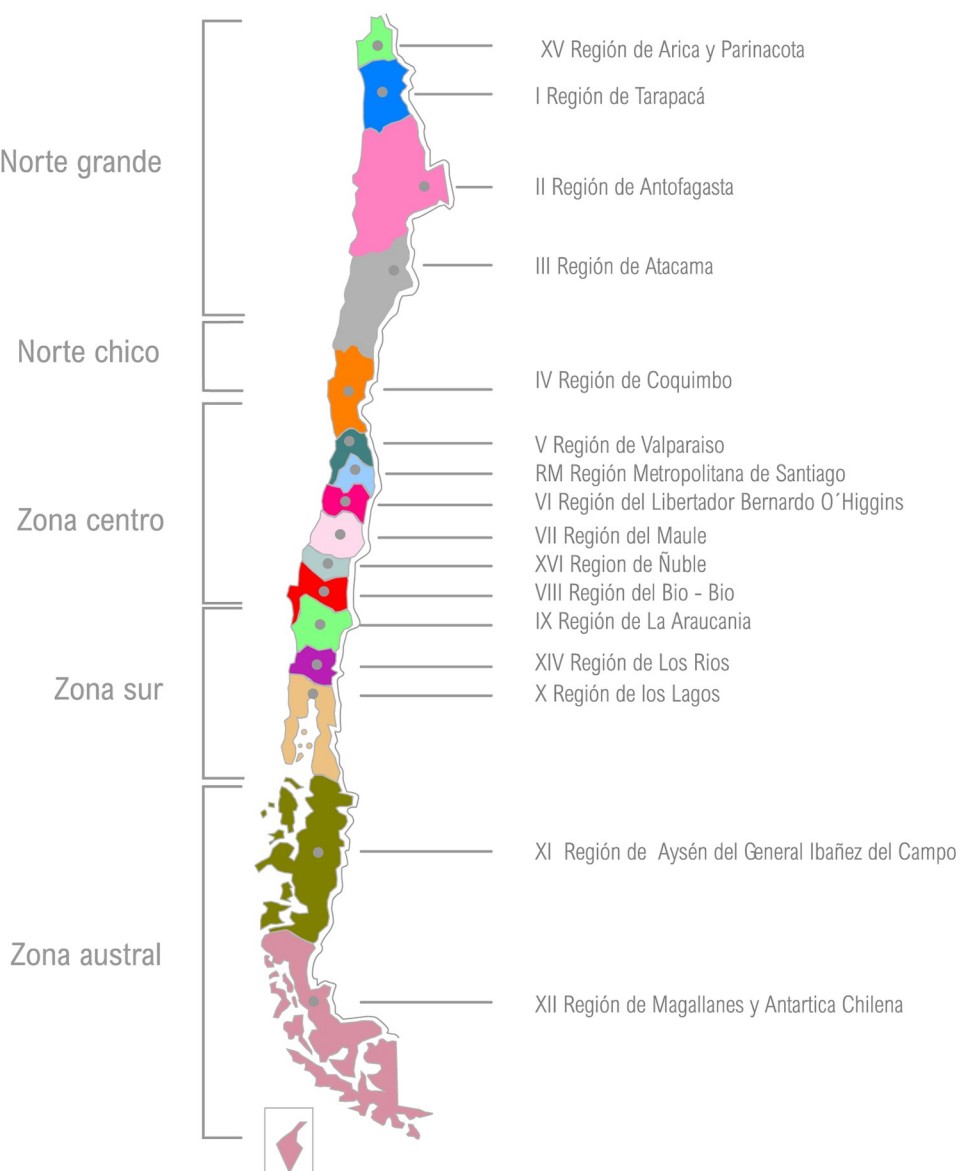

**Fig 1. Political map of Chile showing its administrative and geographical regions.**

and where the outcome was a live newborn. Lastly, they must have signed an informed consent form upon agreeing to participate. A total of 43 participants were selected based on these criteria.

However, bearing in mind that information quality is more important than quantity in qualitative research, feasibility criteria were also applied to this research project [33]. The ability to be in contact with participants without significant difficulties due to large distances across the country of Chile was consequently prioritised. Care was taken not to centralise participant selection in the capital city of Santiago.

Table 1. Participant characteristic.

| Participants | Age | Children previous | Regions of Chile | Partner's presence in childbirth | Home births previous |
|---|---|---|---|---|---|
| P:1 | 36 | 2 | Central Chile | Yes | 0 |
| P:2 | 34 | 1 | Central Chile | Yes | 0 |
| P:3 | 38 | 1 | Central Chile | Yes | 0 |
| P:4 | 36 | 2 | Central Chile | Yes | 0 |
| P:5 | 35 | 2 | Central Chile | Yes | 0 |
| P:6 | 49 | 3 | Central Chile | Yes | 1 |
| P:7 | 33 | 3 | Southern Zone | Yes | 1 |
| P:8 | 34 | 2 | Central Chile | Yes | 0 |
| P:9 | 38 | 2 | Central Chile | Yes | 0 |
| P:10 | 39 | 2 | Central Chile | Yes | 0 |
| P:11 | 32 | 1 | Central Chile | Yes | 0 |
| P:12 | 29 | 3 | Central Chile | Yes | 1 |
| P:13 | 31 | 2 | Central Chile | Yes | 1 |
| P:14 | 34 | 1 | Central Chile | Yes | 0 |
| P:15 | 30 | 1 | Central Chile | No | 0 |
| P:16 | 31 | 2 | Central Chile | Yes | 0 |
| P:17 | 31 | 1 | Central Chile | Yes | 0 |
| P:18 | 32 | 2 | Far North | Yes | 0 |
| P:19 | 30 | 3 | Southern Zone | Yes | 1 |
| P:20 | 40 | 2 | Central Chile | Yes | 0 |
| P:21 | 34 | 2 | Southern Zone | Yes | 0 |
| P:22 | 33 | 1 | Southern Zone | Yes | 0 |
| P:23 | 31 | 2 | Southern Zone | Yes | 0 |
| P:24 | 25 | 3 | Southern Zone | Yes | 0 |
| P:25 | 33 | 2 | Southern Zone | Yes | 0 |
| P:26 | 34 | 3 | Far North | Yes | 0 |
| P:27 | 31 | 2 | Far North | Yes | 1 |
| P:28 | 31 | 2 | Far North | Yes | 0 |
| P:29 | 28 | 1 | Far North | Yes | 0 |
| P:30 | 35 | 3 | Southern Zone | Yes | 1 |

Table by the authors.

All of the above criteria having been applied, 30 participants with the following characteristics were selected in the end (Table 1).

## 5. Data collection method

Data was collected through semi-structured interviews because this technique enabled researchers to 'explore [the interviewee's] life, getting deeper into transcendent details, decoding and comprehending what they like and what they fear, what satisfies and distresses them, and what makes them happy, as significant and relevant to the interview' [34].

A script of interview questions was generated through two complementary processes. The first process entailed a review of the literature and scientific evidence on the topic of home births; this review helped structure the research project goals and identify topics of interest.

The subsequent second process complemented the first through the selection and categorisation of main themes to be addressed. Lastly, questions were drafted in a script used to guide participant interviews (Fig 2).

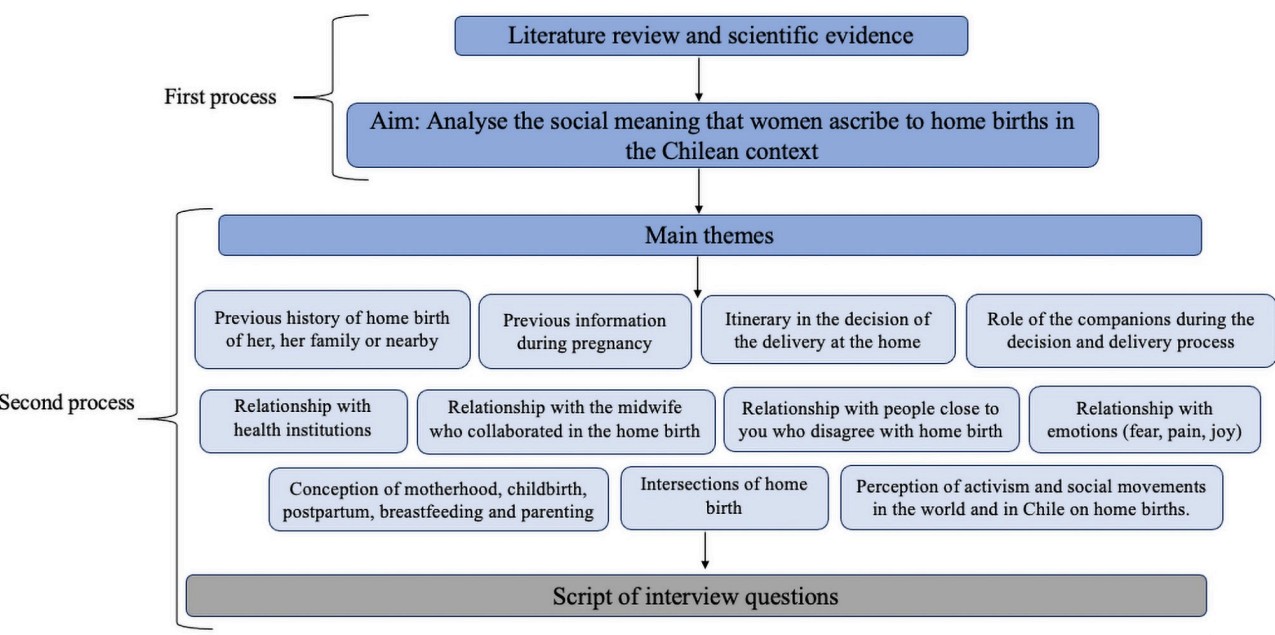

Figure of own elaboration

**Fig 2. Script drafting process.**

This development process resulted in the definitive script used in the interviews (S1 Appendix). All 30 interviews were conducted wherever was convenient for the participants.

The majority took place in participants' homes because most were still nursing.

All interviews were carried out in-person by the manuscript's first author during a single session taking place between the month of October 2018 and January 2019 (Table 2).

## 6. Ethical considerations

This research project was approved by the University of Barcelona bioethics commission under number IRB00003099. The study's aims and ethical considerations were explained by the manuscript's first author to all participants via e-mail and during telephone interviews. As a last step, the participants who were interviewed signed informed consent forms just before taking part in the in-person interview.

It should be noted that this research was supervised by and received administrative support from the Master's programme on gender studies and psychosocial intervention at Universidad Central de Chile as an empirical study carried out in the country of Chile.

## 7. Methodological criteria

The Standards for Reporting Qualitative Research (SRQR) checklist [35] was applied throughout the research process and while drafting the final report. Furthermore, the following criteria by Calderón [36] were adopted: a) 'epistemological and methodological adequacy', by refining the structure of the research question and reviewing process coherency; b) 'relevance', because we clearly and justifiably need to understand the social meaning of home birth for women in terms of their knowledge level. Firstly, this is how we will be able to understand the journey they face. Secondly, this will generate situated knowledge for dissemination in scientific circles.

**Table 2. Interview timetable.**

| Participants | Time elapsed between birth and interview | Number of interview sessions | Date of interview |
|---|---|---|---|
| P:1 | 1 year 4 months | 1 | October 2018 |
| P:2 | 3 years 10 months | 1 | October 2018 |
| P:3 | 1 year 5 months | 1 | October 2018 |
| P:4 | 1 year 3 months | 1 | October 2018 |
| P:5 | 9 months | 1 | October 2018 |
| P:6 | 13 years | 1 | October 2018 |
| P:7 | 1 month | 1 | October 2018 |
| P:8 | 1 year 5 months | 1 | October 2018 |
| P:9 | 2 years | 1 | October 2018 |
| P:10 | 6 months | 1 | October 2018 |
| P:11 | 1 year 10 months | 1 | October 2018 |
| P:12 | 12 months | 1 | October 2018 |
| P:13 | 7 months | 1 | October 2018 |
| P:14 | 11 months | 1 | October 2018 |
| P:15 | 10 months | 1 | October 2018 |
| P:16 | 2 years | 1 | October 2018 |
| P:17 | 3 years | 1 | October 2018 |
| P:18 | 11 months | 1 | November 2018 |
| P:19 | 9 months | 1 | November 2018 |
| P:20 | 3 years 6 months | 1 | December 2018 |
| P:21 | 2 years | 1 | January 2019 |
| P:22 | 10 months | 1 | December 2018 |
| P:23 | 2 years | 1 | November 2018 |
| P:24 | 1 year 6 months | 1 | November 2018 |
| P:25 | 12 months | 1 | November 2018 |
| P:26 | 1 year 6 months | 1 | November 2018 |
| P:27 | 1 year 5 months | 1 | November 2018 |
| P:28 | 12 months | 1 | November 2018 |
| P:29 | 12 months | 1 | November 2018 |
| P:30 | 5 months | 1 | November 2018 |

Table by the authors.

Likewise, the criteria of c) 'validity' should not be understood as a statistical concept, but in terms of pertinency and interpretivism. With these criteria, we ensured that our suitable participant selection process and rigorous analysis would support the pursuit of finding meaning and seeking in-depth explanations that could be generalised using logic and transferred according to the research project's context and circumstances.

Lastly, d) 'reflexivity' is important, having recognised the researchers' feminist perspective as an attempt to affect policy insofar as they have fostered settings in which women who, silenced until now by formal birthing institutions, were able to express themselves. In this sense, using theoretical perspectives that originated in human behavioural and social sciences enabled us to widen and deepen our analysis and understanding of the social meanings that Chilean women ascribe to home birth. It should be noted that this study describes the main findings of a research project associated with a doctoral thesis pursued at the University of Barcelona (Spain) by the first author and supervised by the manuscript's second author.

The first author, a Chilean midwife, was the one to approach to participants initially and interview them in-person. The study's reflexivity criteria reside within the importance bestowed upon recognising the influence that the researcher, the first author in this case, had on the women interviewed instead of trying to prevent that influence. For this reason, the potential for resulting bias arises from the non-explicitation of the research project's reflexivity criteria and not from its inevitable presence.

## 8. Data analysis

Interviews were recorded using a digital recorder. The recordings were subsequently transcribed by the first study author and two external research collaborators. Later, the transcriptions were run through Spanish-language Atlas.ti 8.4.0 software on a Mac computer.

Interview data underwent qualitative thematic analysis entailing the six phases proposed by Braun et al. [37].

During the first phase, the authors 'familiaris [ed themselves] with [their] data'. In this phase, the interviews were transcribed word for word. Next, the transcriptions were read and re-read in order to obtain general ideas from the data provided during the interviews.

The second phase consisted of 'generating initial codes'. This means that the most relevant concepts recovered from the 30 interviews were generally coded and initially organised into meaningful groups, thus preparing them for the next step.

The third phase involved 'searching for themes'. At that point, all pertinent data for analysis were encoded, then classified into general themes. Next, they were grouped and collated in order from general to specific.

The fourth phase consisted of 'reviewing themes'. This stage was divided into two sub-phases. In the first, all excerpts selected for each category were read and researchers checked for category coherence. In the second, they ascertained whether the codes created belonged in or were associated with the category to which they were assigned.

The fifth phase entailed 'defining and naming themes'. In this stage, the main analysis categories were renamed to coherently align them with the codes they contained. Likewise, the researchers made sure that categories and interview excerpts told the 'story' in relation to one another in a way that supported the study objective.

Lastly, in the sixth phase, called 'producing the report', the codes were transformed using the analysis categories described here below into a narrative of the results.

## Results

The average age of the women interviewed was 33.5 years. Seventeen (17) women lived in Central Chile, while eight (8) were in the Southern Zone and five (5) in the Far North.

Most women had more than one child. However, only seven (7) had given birth at home on a prior occasion. All of them had given birth in their own homes, where they were attended to by a midwife an their partners, except for one participant, who did not have a partner at that time.

Study categories were created based on the above information: Main categories: 1) Home birth journeys. Subdivided into two sub-categories: 1.1) Making the decision to give birth at home and 1.2) Giving birth: (re)birth. And secondarily, into four sub-categories: 1.1.1) Why do I need to give birth at home? 1.1.2) The people around me don't support me: obstacles to home birth; 1.2.1) Shifting emotions during home birth, 1.2.2) I (don't) want to be alone (Fig 3).

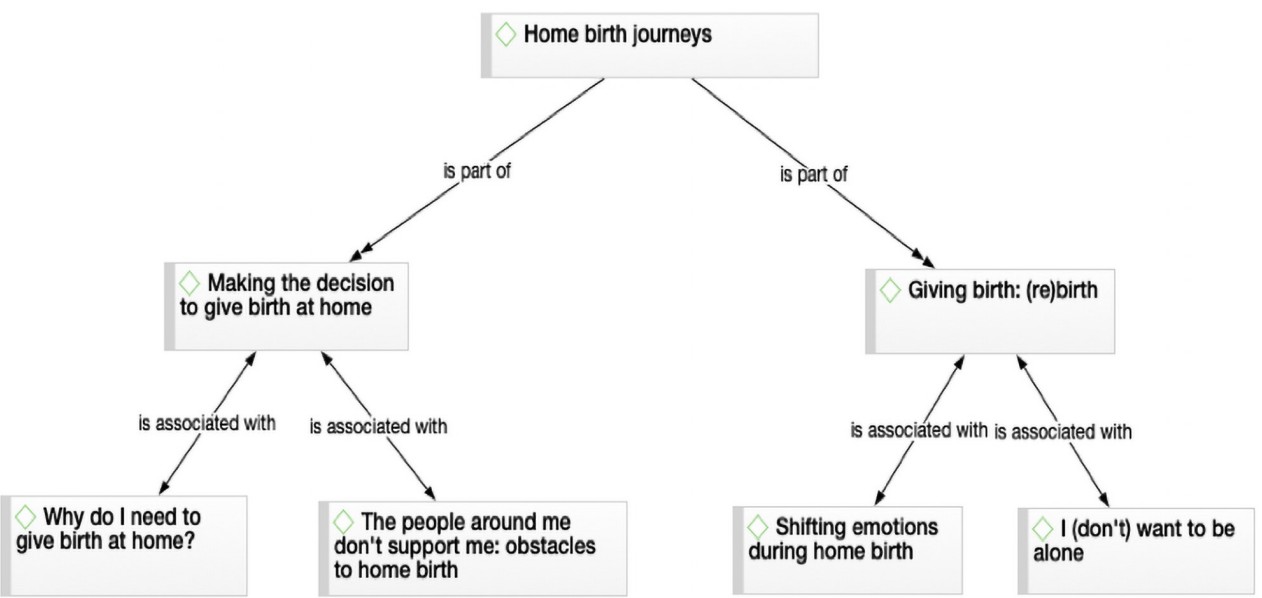

Figure from Atlas ti software

**Fig 3. Categories for analysis.**

## 1. Home birth journeys

The women who participated in this study narrated a series of events that took place from the moment they decided to give birth at home until they actually did so. These events included generating a specific idea about what a home birth would be like, dialogue about the decision with their significant others, telling their families, searching for a midwife to attend to them, and even how they described their connections with healthcare institutions and what the latter represented in their lives.

Home birth journeys are diverse in that they are unique glimpses into the course of a woman's life. Nevertheless, their journey narratives intersect along lines such as obstacles imposed by the healthcare system. On the other hand, they share positive experiences such as their partners' support and the personal satisfaction they found in giving birth at home.

Accordingly, we call these series of events 'home birth journeys' because they are part of each individual's life story, or life journey. As Carmen Lera et al. [38] explains, journeys are the dynamic transitions, experiences and circumstances that allow individuals to structure their own lives for themselves and with others.

**1.1. Making the decision to give birth at home.** According to the women interviewed, one of their hardest moments was making the decision to give birth at home, mainly due to lack of knowledge, or more frequently, the negative ideas surrounding home birth. Social representations of home birth are predominately negative. They are based on claims that home births are unsafe and unsuitable, and therefore women who give birth at home are correspondingly categorised as irresponsible women. One woman expressed it as follows in an interview:

'My mum went crazy. She said that, if it was about the money, she would pay for the birth, and how could I do that, and that I was irresponsible'.

(P: 6:7).

As a source of objective knowledge validated by science, biology and medicine, hygienist discourse has positioned and represented home birth as an unsafe form of childbirth because it is carried out outside of the safeguards of a healthcare institution. This idea has taken deep root in society's conceptual framework of childbirth despite and separately from the vast international scientific evidence that exist on the subject. Women who choose to give birth at home feel obliged to inform themselves about its advantages and disadvantages, both for their own sakes and to generate socially acceptability in their personal surroundings. A participant described it this way:

'It terrified them all, but nobody asked me anything either, except for my dad, who took me aside and said, "Alright, so you want to have [the baby] at home. I want you to explain everything you are doing to have [the baby] at home, what the pros and cons are, what the risks are. . ." so I explained everything to him, and he said "Ah, OK", and relaxed about it'

(P:2:21).

Society's entrenched hygienist discourse creates a situation wherein women who decide to give birth at home have to educate themselves about this option, which entails the complex process of acquiring information from a critical perspective that will allow them to problematise the predominant contemporary way of giving birth and subsequently develop a solid discourse with which to defend and justify their decision. To that end, in order to make and reaffirm their decision based on the benefits of home birth, women's strategies include reading, watching documentaries, attending workshops and dialogue with other women who have given birth at home. Some participants described it as follows:

'I started to read, I started to get informed, and I started to collect lots of information on the benefits of a natural birth in a place I knew, where I was protected, and it started to make a lot of sense to me to have my daughter at home'

(P:3:5).

'I started getting together with friends who had given birth at home, so in that way, yes, a feminine structure helps, such as having close female friends who have gone through this, so like getting information by word of mouth, through sharing. On top of that, at the beginning of my first trimester, I did a workshop about home birth with a midwife'

(P:15:4).

While the women interviewed describe this process of acquiring information and educating themselves as necessary, especially during their first pregnancy, some expressed that, after their second time giving birth, they preferred to let their own experiences be their guide and thereby avoid creating unreal expectations for themselves about a 'perfect' birth that didn't reflect the reality a woman might experience. One interviewee described it as follows:

'During my first pregnancy, I read a lot and I collected a lot of information. I also think that by studying so much, I had created this idea of a perfect birth. So, on my second pregnancy,

I said I wasn't going to read or study anything, and I was just going to let it all happen however it was going to'

(P:12:30).

And so, making the decision to give birth at home reflects the successive situations participants must face during their life journeys, from starting to educate themselves to their firm decision and its reaffirmation to others.

*1.1.1. Why do I need to give birth at home*? The biomedical model is currently the predominant model on labouring and childbirth. The unequal power relationships in this healthcare provision format mean that individuals play passive, secondary roles in their own disease and health processes. Furthermore, this model is part of a technocratic paradigm, an important consideration in the dynamics of the healthcare institution culture, wherein the human body is understood as if it functioned like a machine [15]. Through this metaphor, the technocratic paradigm mechanises physiological processes with a series of procedures and technologies that estrange the idea that women have an autonomous capacity for giving birth. The women interviewed were referring to this situation when they described the excessive medicalisation of births that takes place inside hospitals. From the moment they are admitted, a complete array of technological apparatuses is deployed for application during labour. One participant described it as follows:

'It's quite a nasty situation, a Machiavellian business, medicine is, if you think about, for example, medical machinery. But in the gynaecological area, I have the impression that it's all about being a woman. I feel like right there, something happens, like "women's bodies are defective and we're helping out the woman's body"'

(P:6:14).

This statement expressly addresses the question of sex. At the core of the biomedical model, women are seen as having defective bodies that are insufficiently prepared for giving birth on their own [12], so they need outside intervention to have a 'successful' birth, which is defined solely and exclusively as culmination with a live mother and newborn. But, when birth does not go as expected, the biomedical system attributes this to the inability of women's bodies to give birth according to the established standards, thereby blaming women and making them feel as if they had failed to reach their goal of being mothers [39]. In its zeal to compensate for the deficiencies of women's bodies regarding childbirth, the biomedical system justifies practices and procedures including unwarranted episiotomies, directed pushing, the Kristeller manoeuvre and unnecessary caesarean sections in the name of 'saving' the life of the baby to be born because the mother was unable to give birth by herself. A participant described it this way:

'They made me consent to a C-section I didn't want. They induced me at 8 am and at 1 pm, they told me, "Enough, you never dilated, so we're going to do a C-section on you" as if it were my fault: "you didn't dilate and we can't keep waiting on you", like they were blaming it on me'

(P:12:7).

We observed that the main reason that the women interviewed who had given birth before [in a hospital] gave for wanting home births was negative hospital experiences during their prior pregnancies. Along with ill-treatment and the excessive medicalisation of birth, these

situations have been researched extensively by several different authors [39, 40], who have theorised and conceptualised these acts as 'obstetrical violence', defined by the Venezuelan Fundamental Law on the Right of Women to Live Free of Violence, is 'the appropriation of women's bodies and reproductive processes by healthcare staff' [41]. One participant said:

'I wanted to give birth at home because I was afraid of obstetric violence more than because, "Oh! I wanted to connect with the earth" or something like that. I'm telling you, now that I have given birth at home, if I had another kid, I would do it at home. I recommend it to everyone'

(P: 1:11).

The acts of obstetric violence the participants refer to originate in a system structured such that the rights of women and their families are impinged upon. Multiple factors intersect to cause these manifestations of violence against women in that the women subjected to them coincide in sex and their condition as pregnant individuals; both circumstances place them in vulnerable situations [40].

The experiences the women recount reflect a perinatal trauma that persists in the participants' memories throughout their lives. Despite having had positive experiences with childbirth later on, the emotional and symbolic damage inflicted by perinatal trauma impedes these women from generating and recalling positive feelings about the birth of their children.

A participant described it this way:

'Lots of things happened and a lot of them were really sad. Look, I feel like crying as I'm telling you this right now, even though I later gave birth at home, twice. People say I've already healed from that, and I feel like I won't ever heal because of how they snatched [my child] away from me. They ripped my child from my womb in a violent way. It shouldn't have been like that. And there were a thousand more things, during the C-section, during hard labour and while I was hospitalised. They treated me badly for four days. I was subjected to violence and traumatised so much that I wanted nothing more to do with (the hospital). In fact, I was even scared of going for check-ups'

(P: 12:9).

Thus, opting to give birth at home is a resilience strategy that women adopt in the face of obstetric violence and, simultaneously, an act of emancipation from the current dominant healthcare culture surrounding birth. However, obstacles are not only created within institutional settings. On occasion, they are found among family as well.

*1.1.2. The people around me don't support me*: *obstacles to home birth*. Support and companionship during pregnancy, labour and birth are essential. However, managing their desire to give birth at home, on the one hand, and feeling that they had to persuade their loved ones that this was an informed, safe and responsible decision, on the other, was a significant source of tension for the women interviewed. Sometimes, childbirth outside of institutional regulatory frameworks was interpreted by the families of the women interviewed as an offensively unsafe decision that incurred risk. Families recurred to negative cultural representations such as fear and pain to reverse or question a woman's decision to give birth at home. Biomedical discourse is so entrenched in society's subconscious that it becomes impossible to think about birth in a different, autonomous way that doesn't involve surgery or treatment provided by an institution. One participant described it as follows:

'I felt extremely criticised by my whole family, but not family like my dad, mum and sisters, but my female cousins and my aunts. Actually, I remember a comment one of my cousins made, which I said I wanted to forget because she told me, "I want to see you when you're screaming in pain"'

(P:14:22).

Likewise, giving birth, especially at home, is not a frequent topic of conversation among family members. When someone comments on birth in a family setting, they cautiously and discreetly say that their grandmother or great-grandmother gave birth at home with a birth assistant. Yet rather than being a mere memory or anecdote, these situations intimate that home births are not perceived as a safe model for birthing in the modern age. Technological advances in medicine have given rise to a social discourse that validates and depicts birth in healthcare institutions as the exclusive model, and this rhetoric is intergenerational, as indicated by the following participant:

'My father [. . .] told me he wanted nothing to do with it, that he didn't understand, with all the technology we have, why I would put myself at risk, why I would put my baby's life at risk'

(P:1:14).

In response to these obstacles, some women interviewed chose to hide their decision from their families and thereby avoid complicated interrogations and the possible collapses of family relationships. This selective silence can be understood as a strategy that women use to placate the biopolitical punishment [13] represented by their families as they question and create tension about her decision to give birth outside of the social established norms. A participant described this situation as follows:

'I told my aunt and she said, how could I, with all of the advances we have now, how could I take such a risk? So, I finally told her that OK, I would have [my child] in a hospital and I said the same thing to the whole family, just like I did to my family on my father's side, who are more traditional. I told them my child was born at home afterwards'

(P: 15:18).

On the other hand, the support that women sought was reflected in their partners' active participation. From the very first moment that the interviewees expressed interest in a home birth together with their partners, their partners supported them, although they did have a few questions because they didn't know about labour in a non-institutional context. Partners felt fear and worry about a situation that they neither knew about, nor understood nor even know how to react to. One participant described it as follows:

'I think my husband had a few concerns because he didn't know much about what I wanted to do or what would happen, so we went to check-ups together with the midwife and every time he had questions, I told him, "Ask, ask, so you won't be worried, and you can give me the support I need". So, in the end, we persuaded him'

(P:27:9).

The women interviewed thus described how important their partners' presence was for them while in labour. However, that presence was valued from a household logistics point of view more than for emotional and physical contact. They felt it was much more important for them to have everything under control and play a protective, precautionary role than be together with them during labour. For instance, some interviewees assigned their partners tasks such as keeping the house warm or ensuring they had water or towels on hand. In this sense, Odent [16] comments on a similarity between home births and birth in other mammals, in that the female mammal makes sure she gives birth away from others and the male mammal makes sure that no threats or predators are around. One participant described it this way:

'Before the birth and during labour, he was busy making sure I was alright, like "I need to make sure the house is nice and warm, so she and the baby aren't cold. I need to give her the flowers when she's in pain, I need to massage her. . ."—and maybe I didn't let him give me massages. He made the hole in the ceiling for the cloth. He focused on those issues, resolving material concerns so we could be alright'

(P:14:20).

Despite the family obstacles described by the women interviewed, they experienced the respectful companionship of their partners first as a challenge, and later as support in preparing for the exact time and setting in which they would give birth.

**1.2. Giving birth: (Re)birth.** For the women interviewed, childbirth was not only a continuation of a physiological process but a state that transcended rational comprehension [42]. They experienced emotions that washed over and overwhelmed them as a true re-birth and a sublimation of detachment from worldly concerns. As they gave birth, they felt re-born as women and mothers. One participant described the level of intensity that she experienced as follows:

'It's like you yourself are born again [. . .], when you give birth, you that was before dies, and you are born as a mother, as a powerful woman and that is the most beautiful thing about births. [weeps] I get emotional about it!'

(P: 8:30).

Together with this sensation, the women interviewed expressed that, while in labour, and especially during the more advanced stages, they experienced a trance-like sensation. Some women even compared this sensation to the effects of hallucinogenic substances insofar as sensations of joy, body heat and even a great deal of pleasure. Casilda Rodrigáñez [14] calls this 'orgasmic birth'. She indicates that it 'happens because uterine movement itself produces pleasure, and this happens during natural birth' [14]. She also says that 'the type of movements uterine smooth muscle bundles make during labour is the same as during an orgasm' [14]. Some participants characterised this moment as follows:

'When I wasn't having contractions, I was in a state of pleasure. It was as incredibly intense thing, very sensory and good. I made strange noises and it was like I was in a trance, an exquisitely beautiful trance, like a decadent feeling, very warm. I remember sweating a lot'

(P: 3:13).

'I had taken trips on plants and giving birth is the same. It's like taking some mushrooms or like taking cactus. I don't know what it's called—your sanity, you in your normal state of mind—it's like letting go of that, as if the brain turns off, something like that'

(P: 12:20).

And so, some participants felt sensations and emotions during labour that are totally different from society's negative representations of tragedy and, particularly, pain as a source of suffering. This reveals the possible transition to positive emotions during a home birth.

*1.2.1. Shifting emotions during home birth.* The transcendent understanding beyond rationality as described above is the result of a series of emotions that converge to allow birth to happen. For the women interviewed, home birth was something they had never felt before. They experienced multiple emotions during that time, yet their overall sentiments were satisfaction and joy, so much so that they even recommend home birth to other women. One participant described it as follows:

'Giving birth at home, it's possible. It's beautiful. Beauty isn't always comfortable or easy, but it's worth it, the whole time, and I recommend it to all women. Every woman should try it, really and if they can't, that's OK, but they should try it'

(P:1:48).

The women interviewed describe pain as a primary sensation throughout most of their time labouring, among the multiple emotions they felt throughout the process. The women do not understate the intensity of the pain; in their opinion, it is the most difficult thing to endure. Pain management and the nature of its representation have been the subject of much study [14, 43, 44]. However, the women reinterpreted the pain of childbirth based on the knowledge they had acquired during pregnancy as more closely resembling a necessary emotion for a home birth to take place than as suffering. They describe it as follows:

'It hurts, but you don't suffer. I think that's what you learn from reading and researching until you're sick of reading and researching. At the heart of it, this is not suffering. This is physiological pain, the only time in your life that something is going to hurt because everything is the way it should be'

(P:26:12).

To paraphrase Spinoza [45], no one knows what a body can do, nor what it can achieve. Thus, and contrary to the dominant discourse of 'suffering through birth', emotions such as happiness even coexist with pleasure during home birth despite the pain. These sentiments are a recurring theme among interviewee's stories. Their feelings are made possible because of the fluidity pregnant woman experience giving birth in a place that is familiar and comfortable to them, surrounded by people they have chosen to be there. This is essential.

The more the woman knows and is comfortable with the setting in which she is going to give birth, absent the hostility that occurs in hospitals, the better her body will be able to progress through its physiological process without interruptions or intervention by external agents. In this context, a woman will be able to experience positive emotions and advance through labour and childbirth. As this participant described it:

'Being able to have your baby without someone bothering you, without being interrupted, without anyone getting in the way, and doing it in a private space with your family, being

able to have that memory, for your children (if you have other children and they participate in it) to have that memory: the arrival of a sibling. That's priceless'

(P: 7:39).

During a home birth, women embark on a journey through diverse emotions, mainly experiencing birth as satisfaction, confidence and wisdom. However, sometimes they prefer to experience such an important moment in their lives in introspective solitude over having company.

*1.2.2. I (don't) want to be alone.* The intensity with which emotions appeared during labour was apparent in the women's contradicting and imperative desires to be alone or to have someone else present. This feeling varied among birthing stages. During the initial phases, the prevailing sentiment was the desire for company. Participants expressed that the most meaningful companion for them during that stage was their partner, even if the latter didn't always act the way the women expected them to. Participants felt that their partners transmitted a sense of protection and support in addition to them taking care of home birth logistics. One participant described it this way:

'He came in all of a sudden, all worked up. It annoyed me, but at the same time, I wanted him there. I couldn't imagine him not being there. I had always wanted him to be there during the birth'

(P: 11:9).

Contrarily, during more advanced, active stages of labour, participants expressed having been more comfortable alone, with no one touching or talking to them. It is very important to respect this evolution and refrain from intervening in any way. At this point, the woman and her bodily state play the most important role; any external interposition could impede labour. One participant described it this way:

'The first two hours, I wanted to be alone. I told my husband, "Leave me alone". Like horses go off by themselves, I went to a room and I hugged a cushion. There I was, thinking, alone. During the birth itself, the midwife told him, "Caress her" and I told him, "Don't touch me"!'

(P: 8:8).

Parallel to how important their partners' support was for women, midwives also played an essential role in home birth. The women interviewed said that their midwife's presence transmitted not only a sense of support but also, and in particular, a sense of safety and protection. They perceived the midwife as a professional technical expert and trusted fully in their midwives' knowledge and skills, above all, to know what to do in the event of the unexpected. In the words of one interviewee:

'I needed the midwife to be there to know that everything was going to go well and that if something didn't go well, she would know what to do. She fulfilled that role and took care of my health and my baby's health perfectly'

(P: 5:7).

Midwives played this important role in home births. The women interviewed assessed that their presence meant added value in that, through their attitude of compassion and sisterhood,

the midwives represented feelings that are mutually expressed among women who help other women to give birth. One woman described herself in these words:

> 'I had the feeling that I preferred to be with another woman who would understand the process and what was happening to me better, a woman who could help me'

> (P: 9:12).

In this way, the women interviewed recognise the midwife not only an expert woman with technical and scientific knowledge, but also, they identify a true active partner in the birth process. Their company, understanding and contentment was fundamental, both for them and for their partners and families. Consequently, the figure of the midwife in home birth represents the political and active sisterhood that women need and that challenges midwives to change the world for more just, dignified and humane births [46].

## Discussion

Study participants said that most meaningful and difficult moments in their home birth journeys were making that decision and later carrying it out. This difficulty was mainly associated with the negative discourses that are part of social representations of home births.

As described by Da Rocha et al. [47] childbirth is a sociocultural construction associated to a physiological event, and as such, the pain and negative emotions associated with this construction are seen as intrinsic parts of childbirth.

Furthermore, institutionalised birth is hegemonically understood as the only current model for childbirth. In this regard, Sanfelice et al. [48], Cheyney [49], and Gurara et al. [50] indicate that the concepts of safety and risk have been crucial to overcoming women's fears and uncertainty when they want to give birth outside the context of a healthcare institution, since biomedical discourse justifies its own position that home births entail mortality risks.

Moreover, some authors [51–53] believe that institutionalising birth greatly reduced world maternal and neonatal mortality rates.

Notwithstanding, a Cochrane review performed by Olsen et al. [54] claimed that 'there is no strong evidence from randomised trials to favour either planned hospital birth or planned home birth for low-risk pregnant women' [54]. Hutton et al. [55] came to similar conclusions in their study comparing neonatal and perinatal mortality among women who had planned home births with women with low-risk pregnancies who had planned hospital births.

Arguing that home births should not be performed because maternal and infant mortality and morbidity rates have decreased with the medicalisation of birth restricts the focus of analysis to a debate that solely concerns healthcare, thus overlooking the economic, social, political and demographic variables that affect maternal and infant health throughout the world [17, 56, 57].

Significant international scientific evidence supports birthing in the home. However, healthcare professionals still hesitate to recommend it as a safe option. In this regard, Foucault [13] proposes the concepts of 'biopower' and 'biopolitics' to analyse the new power and control mechanisms that regulate how life and birth are managed and perpetuated in public policy and endeavours. Clavero et al. [58] assert that the technology applied to birth (caesarean sections and the use of forceps, among others) is useful insofar as it has helped human development evolve, and that therefore, it would be very difficult for women to give birth if this technology didn't exist. In short, healthcare professional reticence is founded in not just the discourse on (lack of) home birth safety, as explained by Authors [59], but also stems from a patriarchal dominant structure that conditions women's behaviour and reproductive health.

In addition to the foregoing resistance, the women interviewed express having experienced obstacles within their families. In this regard, they describe having undertaken a significant learning process that comprised defining the problem, reflecting on their motives and finally, generating opposing action [60], which thereby enabled them to state their case and defend their decision to others. Despite these obstacles, the participants felt supported and protected by their partners. Moshi et al. [61] state that a partner's support is important at an emotional level and in a protective sense as they ensure that everything is alright during labour.

With regard to emotions felt during home birth, the women interviewed identified pain as well as joy and pleasure as their principal feelings during labour. Davis-Floyd [15] interprets childbirth as a sexual act and asserts that a woman's privacy should therefore be protected while she is in labour so as to prevent a mental and hormonal disconnect. This relates directly to the interviewees' desire for company or, frequently, to be alone during labour. Odent [16] attributes this contradictory phenomenon to neocortical inhibition. This author states that the neocortex must stop working during labour because, it might inhibit childbirth. Because this part of the brain is the agent responsible for rational activity, any stimulus that would activate it should be avoided.

Participants stated that, during labour, they felt calm and confident because they knew there was a midwife with them. Some authors [62–66] emphasise the importance of being attended by a qualified midwife during a home birth for the woman to feel confident and safe, especially in light of the possible complications. Cheyney et al. [67] state that adequate communication between the home and the hospital is important in case a transfer is required.

The limitations of this study include the absence of the stories of women who gave birth at home without a midwife or in the company of a traditional birth assistant or doula. These situations would be tremendously interesting for future research to deepen our understanding of the social meanings of home births. They were not incorporated in this research project due to feasibility concerns.

## Conclusion

Analysing the social meanings that women ascribe to home birth in Chile entailed an intense review that revealed, on the one hand, the historical omission of women in these matters by patriarchal and colonising agents, and on the other, how difficult home birth is made due to healthcare, political and legislative issues. Identifying these tensions because is important because it will enable us to create and develop a new and complementary culture of birth that is specific to home birthing and that would enable alternative forms and community perspectives of motherhood, thereby opening up paths to propose and defend the healthcare ethics of labour and childbirth. Recognising the knowledge and wisdom of women's accumulated life experience enables births, motherhood and child-raising to be included within feminist and popular movements.

In conclusion, this study reveals how urgently necessary it is to defend and make home birth visible and recognised as a safe and valid option, and above all, to create a policy for women's well-being in response to a dominant hegemonic system that disenfranchises women and disables their decision-making.

## Supporting information

**S1 Appendix. Script of interview questions.**
(TIFF)

## Acknowledgments

We are grateful, first and foremost, to the Chilean women interviewed who generously participated and shared their childbirth stories as part of this study.

We would also like to thank the Master's programme in gender studies and psychosocial intervention at Universidad Central de Chile for collaborating on and supervising the empirical work carried out in Chile.

Additional thanks to Proof Editing SL and the translator collaborating with them for their linguistic support on the English version of the manuscript.

This paper contributes to the EU COST Action CA18211 "DEVOTION: Perinatal Mental Health and Birth Related Trauma: Maximising best practice and optimal outcomes".

## Author Contributions

**Conceptualization:** Pía Rodríguez-Garrido, Josefina Goberna-Tricas.

**Data curation:** Pía Rodríguez-Garrido.

**Formal analysis:** Pía Rodríguez-Garrido, Josefina Goberna-Tricas.

**Funding acquisition:** Pía Rodríguez-Garrido, Josefina Goberna-Tricas.

**Investigation:** Pía Rodríguez-Garrido, Josefina Goberna-Tricas.

**Methodology:** Pía Rodríguez-Garrido, Josefina Goberna-Tricas.

**Project administration:** Pía Rodríguez-Garrido, Josefina Goberna-Tricas.

**Resources:** Pía Rodríguez-Garrido, Josefina Goberna-Tricas.

**Software:** Pía Rodríguez-Garrido, Josefina Goberna-Tricas.

**Supervision:** Josefina Goberna-Tricas.

**Validation:** Pía Rodríguez-Garrido, Josefina Goberna-Tricas.

**Visualization:** Pía Rodríguez-Garrido, Josefina Goberna-Tricas.

**Writing – original draft:** Pía Rodríguez-Garrido, Josefina Goberna-Tricas.

**Writing – review & editing:** Pía Rodríguez-Garrido, Josefina Goberna-Tricas.

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
