## [Decision Letter · Decision Letter 0]

15 Dec 2020

PONE-D-20-29567

Birth Cultures: Home Births in the Chilean Context

PLOS ONE

Dear Dr. Goberna-Tricas,

Thank you for submitting your manuscript to PLOS ONE. After careful consideration, we feel that it has merit but does not fully meet PLOS ONE’s publication criteria as it currently stands. Therefore, we invite you to submit a revised version of the manuscript that addresses the points raised during the review process.

The manuscript and the reviewers’ comments were carefully evaluated. The manuscript was appreciated by the Reviewers. Nevertheless, as suggested, the manuscript requires extensive improvement before to be considered for publication. Suggested revisions are in detail reported in the Reviewers’ comments.

We look forward to receiving your revised manuscript.

Kind regards,

Simone Garzon

Academic Editor

PLOS ONE

Journal Requirements:

2.) Please include your tables as part of your main manuscript and remove the individual files. Please note that supplementary tables (should remain/ be uploaded) as separate "supporting information" files

3.) Please include additional information regarding the survey or questionnaire used in the study and ensure that you have provided sufficient details that others could replicate the analyses. For instance, if you developed a questionnaire as part of this study and it is not under a copyright more restrictive than CC-BY, please include a copy, in the original language as well as the English version already provided, as Supporting Information.

4.) During our internal checks, the in-house editorial staff noted that you conducted research in another country. Please ensure that you have suitably acknowledged the contributions of any local collaborators involved in this work in your authorship list and/or Acknowledgements. Authorship criteria is based on the International Committee of Medical Journal Editors (ICMJE) Uniform Requirements for Manuscripts Submitted to Biomedical Journals - for further information please see here: https://journals.plos.org/plosone/s/authorship.

5.) In your Methods section, please provide additional information about the participant recruitment method and the demographic details of your participants. Please ensure you have provided sufficient details to replicate the analyses such as: a) the recruitment date range (month and year), b) a statement as to whether your sample can be considered representative of a larger population, and c) descriptions of where the research took place.

6.) We note that Figure 1 in your submission contains map images which may be copyrighted. All PLOS content is published under the Creative Commons Attribution License (CC BY 4.0), which means that the manuscript, images, and Supporting Information files will be freely available online, and any third party is permitted to access, download, copy, distribute, and use these materials in any way, even commercially, with proper attribution. For these reasons, we cannot publish previously copyrighted maps or satellite images created using proprietary data, such as Google software (Google Maps, Street View, and Earth). For more information, see our copyright guidelines: http://journals.plos.org/plosone/s/licenses-and-copyright.

7) Please amend either the title on the online submission form (via Edit Submission) or the title in the manuscript so that they are identical.

8) Please include captions for your Supporting Information files at the end of your manuscript, and update any in-text citations to match accordingly. Please see our Supporting Information guidelines for more information: http://journals.plos.org/plosone/s/supporting-information. 

Reviewers' comments:

Reviewer's Responses to Questions

**Comments to the Author**

1. Is the manuscript technically sound, and do the data support the conclusions?

Reviewer #1: Yes

Reviewer #2: Yes

Reviewer #3: Yes

2. Has the statistical analysis been performed appropriately and rigorously? 

Reviewer #1: N/A

Reviewer #2: N/A

Reviewer #3: N/A

3. Have the authors made all data underlying the findings in their manuscript fully available?

Reviewer #1: Yes

Reviewer #2: Yes

Reviewer #3: Yes

4. Is the manuscript presented in an intelligible fashion and written in standard English?

Reviewer #1: Yes

Reviewer #2: Yes

Reviewer #3: Yes

5. Review Comments to the Author

Reviewer #1: I really value what you have done.

One concern I have is that you seem to accept the claim that medicine has made that hospitalization is responsible for the lowered maternal and infant morbidity and mortality rates historically, and the data does not support that. Over time, parity decreased, age at first births increased, some general public health measures occurred -- all of which improved infant and maternal health. Where we actually have the data we have been able to see that hospitalization and the destruction of midwifery care harmed not improved outcomes.

That was my main concern.

Reviewer #2: Thanks for giving me the opportunity to review this study which uses qualitative methods to explore the social meaning of home births perceived by women giving birth in several regions of Chile. Below some comments into the different sections:

Title

As SRQR guidelines, it would be useful to include the qualitative approach in the title.

Abstract:

An abstract is included with clear aim, methods. Enumeration of categories/subcategories in “Findings” seems a bit confusing. Was there one major theme “home birth journey” with two subthemes: “making the decision to give birth at home” (Why do I need to give birth at home?; the people around me don’t support me), and “giving birth” (Shifting emotions during home birth; I (don’t) want to be alone).

Introduction

The introduction provides a good overview of birth cultures, feminists approach to motherhood and rational for research. Some citations need to be amended to Vancouver style.

Methods

The methods section describes a qualitative approach (using a phenomenological paradigm and feminist perspective) and rational for this. Context, setting, participant selection is generally well described. Perhaps more contextual factors could be described (e.g. maternity health system, general population health, needs). Recruitment process is not very detailed (e.g. written info provided? who gained consent, timeline, how many weeks after birth were women interviewed? how participants were selected among the 100 interested?). Why were those baseline characteristics selected?

The approval ethical statement needs to be updated in the methods section (it is XXX at the moment). In the submission form authors mention ethical approval from the University of Barcelona, Spain. Since this study was conducted in Chile, it is surprising not to have any Chilean regulatory ethical body review or approval. No author has any Chilean affiliation and no Chilean organisation is acknowledged. Could this have any implications for research and contextualisation of findings? It would be useful for authors to clarify this.

In terms of data collection: How were interviews conducted (face to face, online?)? Who did the interviews/researcher’s characteristics and reflexivity? Data analysis and different steps of thematic analysis are very well described. It would be useful to know if any specific techniques were used to guarantee rigour/trustworthiness of data.

Results/Findings

Having here the table with characteristics of interviews of women will be very useful, as well as the timeline for interviews (X days/weeks/months after birth). A summary of 1-2 lines of baseline characteristics will be useful.

It feels that some statements in these sections are not findings from this study explicitly and should perhaps fit better within the discussion section and referenced (e.g. page 15: The infamy surrounding home births is no coincidence. Let us recall that in olden days, women gave birth inside their homes, attended to by a birth assistant. However, the modernisation ushered in by the industrial revolution has resulted in the institutionalisation of birth…..; and women have gone from labouring in the privacy of their homes to inside hospitals. This has led to a lower maternal and neonatal mortality rate;…; page 19: “Obstetric violence encompasses a series of practices and attitudes such as physical aggression toward pregnant women, verbal abuse, acts of coercion, infantilisation and the performance of unjustified caesarean sections…” ; page 22: “The instant the baby is pushed out during a vaginal birth is regarded as a culminating moment, the climax of a series of stages that, from a physiological point of view, correspond to neurohormonal and physical events”; page 24: as stated by Le Breton [37], emotions are ‘a real or imaginary, past, present….etc)

Discussion is well presented with links to empirical data and good refs (again citations and ref need to be updated to Vancouver style). P28. “It is evident that significant scientific evidence exists supporting home birth. However, healthcare professionals still hesitate to recommend it as a safe option” Does this refer to Chile? It feels this is hugely dependant on context. Are there limitations of the study that need to be considered? (e.g. sample size, lack of data on baseline socio-demographic characteristics)

Reviewer #3: This paper describes a well-done study and insightful analysis, which contributes to the understanding of social importance and experience of childbirth in the Chilean context.

The following changes changes would make the paper stronger and easier for the reader assess the merits of the study:

INTRODUCTION

This study has a strong theoretical foundation, but the authors don't make a clear case for the need for the current research. What gap in knowledge is it filling?

Minor note: Robbie Davis-Floyd is a woman. The sentence citing reference 15 should read "for her part..."

There are also a few sentences in the introduction that are not clear:

The dominant structures

crossed by economic and political models that guide the behaviors of society, have

precarious and violated the collective dynamics surrounding motherhood and parenting (Missing a word?)

Situation that worsens in the puerperium with breastfeeding [13] (Incomplete sentence)

However, not all advocates are homogenous. (Do you mean not all contexts are homogeneous?)

METHODS

Can you provide more justification for the geographical diversity of the sample, and how you expected this to affect your results? The availability of health care is very different across the three sample regions. Did you expect the themes to be different?

Please provide more detail regarding sampling and inclusion criteria. You state that you had 110 potential participants, but you only interviewed 30. Were any of them excluded for reasons other than the inclusion criteria?

Also, why 30 participants? Did you think you would reach saturation with 30? Did you actually reach saturation? Was 30 selected for practical considerations?

Please provide more description in the text regarding the two "complementary processes" involved in the interview script development. These processes are not clear. From the figure, it looks like you identified main themes from a literature search-- that's one process. What was the second?

RESULTS

There are several paragraphs in the results section that are more appropriate for the discussion. The results section should describe your findings, not describe related literature.

DISCUSSION AND CONCLUSIONS

These were strong, but could be enhanced by incorporating some of the content that was prematurely included in the results.

Lastly, you state that the "data are available," but do not note how and where readers could access this data. This should be remedied before publication.

6. PLOS authors have the option to publish the peer review history of their article (what does this mean?). If published, this will include your full peer review and any attached files.

Reviewer #1: **Yes: **Barbara Katz Rothman

Reviewer #2: No

Reviewer #3: **Yes: **Fiona H Weeks

---

## [Author Response · Author response to Decision Letter 0]

29 Jan 2021

Academic Editor

Authors

The rebuttal letter has been completed and the modifications suggested by the Academic Editor and the reviewers have been implemented. It is attached as “Response to Reviewers”.

The final version is attached, entitled “Revised Manuscript with Track Changes”.

The final version of the file is attached, entitled “Manuscript”.

Journal Requirements

1.Please ensure that your manuscript meets PLOS ONE's style requirements, including those for file naming. 

2.Please include your tables as part of your main manuscript and remove the individual files. Please note that supplementary tables (should remain/ be uploaded) as separate "supporting information" files.

3.Please include additional information regarding the survey or questionnaire used in the study and ensure that you have provided sufficient details that others could replicate the analyses. For instance, if you developed a questionnaire as part of this study and it is not under a copyright more restrictive than CC-BY, please include a copy, in the original language as well as the English version already provided, as Supporting Information.

4.During our internal checks, the in-house editorial staff noted that you conducted research in another country. Please ensure that you have suitably acknowledged the contributions of any local collaborators involved in this work in your authorship list and/or Acknowledgements. 

5.In your Methods section, please provide additional information about the participant recruitment method and the demographic details of your participants. Please ensure you have provided sufficient details to replicate the analyses such as: a) the recruitment date range (month and year), b) a statement as to whether your sample can be considered representative of a larger population, and c) descriptions of where the research took place.

6.We note that Figure 1 in your submission contains map images which may be copyrighted. All PLOS content is published under the Creative Commons Attribution License (CC BY 4.0), which means that the manuscript, images, and Supporting Information files will be freely available online, and any third party is permitted to access, download, copy, distribute, and use these materials in any way, even commercially, with proper attribution. For these reasons, we cannot publish previously copyrighted maps or satellite images created using proprietary data, such as Google software (Google Maps, Street View, and Earth).

7.Please amend either the title on the online submission form (via Edit Submission) or the title in the manuscript so that they are identical

8.Please include captions for your Supporting Information files at the end of your manuscript, and update any in-text citations to match accordingly. 

Authors:

1.Completed

2.Completed

3.No questionnaire was used during the study. However, the interview script is attached as Supporting Information.

4.Completed

5.Completed

6.The author’s signed permission to use her figure is attached. A legend has been included in the figure 1 that mentions said permission and the names the figure’s author.

7.Completed

8.Completed

Reviewer 1

One concern I have is that you seem to accept the claim that medicine has made that hospitalization is responsible for the lowered maternal and infant morbidity and mortality rates historically, and the data does not support that. Over time, parity decreased, age at first births increased, some general public health measures occurred -- all of which improved infant and maternal health. Where we actually have the data, we have been able to see that hospitalization and the destruction of midwifery care harmed not improved outcomes.

That was my main concern.

Authors

The wording of the Results section has been modified (p. 20) and a paragraph has been added to the Discussion section (p. 33) clarifying that lowered maternal and infant mortality rates are not due exclusively to the hospitalization of childbirth.

Reviewer 2

Title:

As SRQR guidelines, it would be useful to include the qualitative approach in the title.

Abstract:

An abstract is included with clear aim, methods. Enumeration of categories/subcategories in “Findings” seems a bit confusing. Was there one major theme “home birth journey” with two subthemes: “making the decision to give birth at home” (Why do I need to give birth at home? the people around me don’t support me), and “giving birth” (Shifting emotions during home birth; I (don’t) want to be alone).

Introduction

The introduction provides a good overview of birth cultures, feminists approach to motherhood and rational for research. Some citations need to be amended to Vancouver style.

Methods

The methods section describes a qualitative approach (using a phenomenological paradigm and feminist perspective) and rational for this. Context, setting, participant selection is generally well described. Perhaps more contextual factors could be described (e.g., maternity health system, general population health, needs). Recruitment process is not very detailed (e.g., written info provided? who gained consent, timeline, how many weeks after birth were women interviewed? how participants were selected among the 100 interested?). Why were those baseline characteristics selected?

The approval ethical statement needs to be updated in the methods section (it is XXX at the moment). In the submission form authors mention ethical approval from the University of Barcelona, Spain. Since this study was conducted in Chile, it is surprising not to have any Chilean regulatory ethical body review or approval. No author has any Chilean affiliation and no Chilean organization is acknowledged. Could this have any implications for research and contextualization of findings? It would be useful for authors to clarify this.

In terms of data collection: How were interviews conducted (face to face, online?)? Who did the interviews/researcher’s characteristics and reflexivity? Data analysis and different steps of thematic analysis are very well described. It would be useful to know if any specific techniques were used to guarantee rigour/trustworthiness of data.

Results/Findings

Having here the table with characteristics of interviews of women will be very useful, as well as the timeline for interviews (X days/weeks/months after birth). A summary of 1-2 lines of baseline characteristics will be useful.

It feels that some statements in these sections are not findings from this study explicitly and should perhaps fit better within the discussion section and referenced (e.g., page 15: The infamy surrounding home births is no coincidence. Let us recall that in olden days, women gave birth inside their homes, attended to by a birth assistant. However, the modernization ushered in by the industrial revolution has resulted in the institutionalisation of birth….; and women have gone from laboring in the privacy of their homes to inside hospitals. This has led to a lower maternal and neonatal mortality rate;…; page 19: “Obstetric violence encompasses a series of practices and attitudes such as physical aggression toward pregnant women, verbal abuse, acts of coercion, infantilisation and the performance of unjustified caesarean sections…” ; page 22: “The instant the baby is pushed out during a vaginal birth is regarded as a culminating moment, the climax of a series of stages that, from a physiological point of view, correspond to neurohormonal and physical events”; page 24: as stated by Le Breton [37], emotions are ‘a real or imaginary, past, present….etc)

Discussion is well presented with links to empirical data and good refs (again citations and ref need to be updated to Vancouver style). P28. “It is evident that significant scientific evidence exists supporting home birth. However, healthcare professionals still hesitate to recommend it as a safe option” Does this refer to Chile? It feels this is hugely dependant on context. Are there limitations of the study that need to be considered? (e.g., sample size, lack of data on baseline socio-demographic characteristics).

Authors:

The qualitative approach has been added to the title.

The explanation of the categories in the Results section has been clarified. 

The entire text has been reviewed to ensure compliance with Vancouver style.

A new subsection, “Healthcare and Birthing in Chile”, has been created. In this section, the social facets of healthcare associated with home births in Chile have been explored.

Participant recruitment and selection criteria have been addressed in greater detail and a table has been added to help explain the interview timeline.

In the Ethical Considerations section, the way in which the first author of the manuscript performed informed consent and in-person interviews has been described.

In the Ethical Considerations section, the Master’s program in gender studies and psychosocial intervention at Universidad Central de Chile has been mentioned as the academic entity that supervised and collaborated on the logistical aspects of empirical work carried out in Chile.

In the Acknowledgements section, said Master’s program has been mentioned again and thanked for their support and collaboration. 

In the Data Collection section, the way in which the first author of the manuscript carried out the in-person interviews has been explained in greater detail.

In the Methodological Rigor section, reflexivity criteria have been described in detail together with the rest of the criteria that shape the methodological rigor of the research performed.

Some participant characteristics have been outlined briefly in the Results section.

The wording and description in the indicated paragraphs have been corrected, and some theoretical reflections that do not correspond to the Results section have been deleted. 

The citations have been corrected to comply with Vancouver style.

On page 33, an explanation that the vast amount of existing scientific evidence concerning home births exists at the international level has been inserted.

At the end of the Discussion section, the study’s main limitations have been stated in detail.

Reviewer 3

Introduction

This study has a strong theoretical foundation, but the authors don't make a clear case for the need for the current research. What gap in knowledge is it filling?

Minor note: Robbie Davis-Floyd is a woman. The sentence citing reference 15 should read "for her part..."

There are also a few sentences in the introduction that are not clear:

The dominant structures

crossed by economic and political models that guide the behaviors of society, have

precarious and violated the collective dynamics surrounding motherhood and parenting (Missing a word?)

Situation that worsens in the puerperium with breastfeeding [13] (Incomplete sentence)

However, not all advocates are homogenous. (Do you mean not all contexts are homogeneous?)

Methods

Can you provide more justification for the geographical diversity of the sample, and how you expected this to affect your results? The availability of health care is very different across the three sample regions. Did you expect the themes to be different?

Please provide more detail regarding sampling and inclusion criteria. You state that you had 110 potential participants, but you only interviewed 30. Were any of them excluded for reasons other than the inclusion criteria?

Also, why 30 participants? Did you think you would reach saturation with 30? Did you actually reach saturation? Was 30 selected for practical considerations?

Please provide more description in the text regarding the two "complementary processes" involved in the interview script development. These processes are not clear. From the figure, it looks like you identified main themes from a literature search-- that's one process. What was the second?

Results

There are several paragraphs in the results section that are more appropriate for the discussion. The results section should describe your findings, not describe related literature.

Discussion and Conclusions

These were strong but could be enhanced by incorporating some of the content that was prematurely included in the results.

Authors:

The justification for addressing this particular problem has been improved, and the contribution that this research makes to the international and national scientific communities has been explained.

The wording of the sentence beginning “For his part…” has been modified.

The wording of the paragraph beginning “The dominant structures…” has been modified.

The sentence beginning “Situation that worsens in…” has been deleted because it did not fit the context.

The explanation of difference among countries as heterogenous contexts has been improved. 

The importance of incorporating women from different regions in Chile into the study has been incorporated. 

The feasibility criteria applied during the qualitative research process as related to sampling and inclusion has been described. The consideration of feasibility and practical aspects of carrying out this research project enabled both the researcher and the interviewee to determine their availability for participating in the study. Relevant aspects included geographical distance, mobility and availability of time (most participants had given birth no more than one year previously).

The explanation of the processes by which the interview questions were drafted has been improved.

Some paragraphs have been deleted and others have been moved to the Discussion section. 

Some of the Discussion section paragraphs have been modified and others from the Results section have been added.

---

## [Decision Letter · Decision Letter 1]

2 Mar 2021

PONE-D-20-29567R1

Birth Cultures: A Qualitative Approach to Home Birthing in Chile

PLOS ONE

Dear Dr. Goberna-Tricas,

Thank you for submitting your manuscript to PLOS ONE. After careful consideration, we feel that it has merit but does not fully meet PLOS ONE’s publication criteria as it currently stands. Therefore, we invite you to submit a revised version of the manuscript that addresses the points raised during the review process.

Please, perform recommended revision by reviewer 3 for final acceptance. 

We look forward to receiving your revised manuscript.

Kind regards,

Simone Garzon

Academic Editor

PLOS ONE

Journal Requirements:

Reviewers' comments:

Reviewer's Responses to Questions

**Comments to the Author**

1. If the authors have adequately addressed your comments raised in a previous round of review and you feel that this manuscript is now acceptable for publication, you may indicate that here to bypass the “Comments to the Author” section, enter your conflict of interest statement in the “Confidential to Editor” section, and submit your "Accept" recommendation.

Reviewer #1: All comments have been addressed

Reviewer #3: All comments have been addressed

2. Is the manuscript technically sound, and do the data support the conclusions?

Reviewer #1: Yes

Reviewer #3: Yes

3. Has the statistical analysis been performed appropriately and rigorously? 

Reviewer #1: N/A

Reviewer #3: N/A

4. Have the authors made all data underlying the findings in their manuscript fully available?

Reviewer #1: Yes

Reviewer #3: Yes

5. Is the manuscript presented in an intelligible fashion and written in standard English?

Reviewer #1: Yes

Reviewer #3: Yes

6. Review Comments to the Author

Reviewer #1: apologies for taking so long. All good. And now filling up space because they want 100 characters....

Reviewer #3: The additions regarding the context and sampling decisions greatly strengthen the paper. I have one small note regarding an addition. This sentence is a bit confusing:

Of those, 33% entailed caesarean sections performed via the public system and 63% took place within

the private system, positioning Chile as a country with one of the world’s highest rates of

C-sectioning [31].

I recommend rephrasing for clarity:

Within the public system, 33% of births are by cesarean delivery and 63% of those in private system are by cesarean, positioning Chile as a country with one of the world’s highest rates of

C-sectioning [31].

7. PLOS authors have the option to publish the peer review history of their article (what does this mean?). If published, this will include your full peer review and any attached files.

Reviewer #1: No

Reviewer #3: **Yes: **Fiona H Weeks

---

## [Author Response · Author response to Decision Letter 1]

4 Mar 2021

Journal Requirements:

-The reference list is complete and correct.

Academic Editor

-Ok

-Ok

-Ok

Reviewer 3:

The additions regarding the context and sampling decisions greatly strengthen the paper. I have one small note regarding an addition. This sentence is a bit confusing:

Of those, 33% entailed caesarean sections performed via the public system and 63% took place within the private system, positioning Chile as a country with one of the world’s highest rates of C-sectioning [31].

I recommend rephrasing for clarity:

Within the public system, 33% of births are by cesarean delivery and 63% of those in private system are by cesarean, positioning Chile as a country with one of the world’s highest rates of C-sectioning [31].

Authors

Completed.

The sentence has been modified according to the suggestion of the Reviewer #3.

We have added this modification suggested by Reviewer 3 in the manuscript that already contained the previous modifications. (See page n.7). Please, tell us if we should previously accept the modifications of the first revision and add "with Track Changes" only this modification suggested in the current revision.

---

## [Editor Report · Decision Letter 2]

15 Mar 2021

Birth Cultures: A Qualitative Approach to Home Birthing in Chile

PONE-D-20-29567R2

Dear Dr. Goberna-Tricas,

We’re pleased to inform you that your manuscript has been judged scientifically suitable for publication and will be formally accepted for publication once it meets all outstanding technical requirements.

Kind regards,

Simone Garzon

Academic Editor

PLOS ONE
---

## [Editor Report · Acceptance letter]

5 Apr 2021

PONE-D-20-29567R2 

Birth Cultures: A qualitative approach to home birthing in Chile 

Dear Dr. Goberna-Tricas:

I'm pleased to inform you that your manuscript has been deemed suitable for publication in PLOS ONE. Congratulations! Your manuscript is now with our production department. 

Kind regards, 

on behalf of

Dr. Simone Garzon 

Academic Editor

PLOS ONE